# Tocotrienols in Different Parts of Wild *Hypericum perforatum* L. Populations in Poland

**DOI:** 10.3390/molecules30051137

**Published:** 2025-03-02

**Authors:** Paweł Górnaś, Edyta Symoniuk

**Affiliations:** 1Institute of Horticulture, Graudu 1, LV-3701 Dobele, Latvia; 2Department of Food Technology and Assessment, Institute of Food Sciences, Warsaw University of Life Sciences, Nowoursynowska 166, 02-787 Warsaw, Poland; edyta_symoniuk@sggw.edu.pl

**Keywords:** *Guttiferaceae*, *Hypericaceae*, herb, phytomedicine, bioactive lipophilic phytochemicals, tocol, vitamin E

## Abstract

St. John’s wort (*Hypericum perforatum* L.) is a well-known medicinal plant widely used in phytotherapy due to its abundant secondary metabolites with bioactive properties. Research on tocochromanols conducted in nine populations of St. John’s wort using reverse-phase high performance liquid chromatography with fluorescent light detector (RP-HPLC-FLD) has provided new insights into their diverse composition in different aerial parts of the plant. Flower buds displayed the most diverse tocochromanol profile, while stems contained the lowest levels of these compounds. In most of the examined tissues, δ-T3 was the predominant tocochromanol (34–69%), except in leaves, where α-T dominated. The highest concentration of total tocotrienols was recorded in flower buds (48.5–74.1 mg/100 g dry weight), with δ-T3 (56%) and α-T3 (38%) as the dominant compounds. The highest δ-T3 content was found in unripe seed pods (29.6–45.1 mg/100 g dw). Principal Component Analysis (PCA) revealed distinct differences in tocochromanol composition among the aerial parts of *H. perforatum*. The findings highlight that *H. perforatum* has higher potential applications in the food, pharmaceutical, and medical industries than previously assumed, due to its relatively high content of tocotrienols, mainly δ-T3, in different plant organs.

## 1. Introduction

*Hypericum perforatum* L., known as St John’s wort, is a flowering plant belonging to the family Hypericaceae. This perennial herb can grow up to 1 m tall and is characterized by its many yellow flowers, which have distinct black dots around their edges. The plant features long stamens and three pistils, contributing to its unique floral structure. It is a perennial herbaceous plant that can be found in various temperate regions across the globe and is native to western Asia, Europe, and northern Africa. Over time, St. John’s wort was introduced to North America, South America, southern Africa, Australia, New Zealand, and Japan. This plant thrives in sunny locations, making it commonly found in dry soils and abandoned fields [1].

St. John’s wort has a long history of use as a medicinal plant. It is widely recognized as a natural antidepressant and is often recommended for managing mild to moderate mood disorders [2]. In folk medicine, St. John’s wort is used to address digestive issues, including bloating, abdominal pain, and inflammation of the stomach lining. Externally, it is applied to treat wounds, burns (including first- and second-degree burns), eczema, frostbite, and slow-healing wounds [3]. Its anti-inflammatory properties make it effective for treating abscesses and ulcers. St. John’s wort is also valued for supporting liver function and promoting bile secretion, which can be beneficial in managing liver disorders [4]. Historically, it was used to relieve neuralgia, neuroses, and other nervous system disorders. In ancient times, it was believed to offer protection against “devilish influences” and was even used to treat snake bites [5]. Additionally, St. John’s wort has been applied to skin conditions such as vitiligo and employed as a gargle for oral infections. In traditional medicine, it was also used to treat hemorrhoids and as a diuretic [6]. Furthermore, the anticancer effect of this plant has also been demonstrated [7].

The different medical properties of St. John’s wort result from its diverse composition of bioactive compounds. It contains, among other compounds, hypericin, hyperforin, hyperoside, flavonoids (e.g., quercetin and rutin), phenolic acids (e.g., caffeic acid and chlorogenic acid), tannins, and essential oils [8]. Hypericin is the main active component of St. John’s wort, and together with hyperforin, it is responsible for the antidepressant properties of this medical plant. Its flowering parts have been utilized in traditional and phytomedicines due to the presence of specific bioactive phytochemicals such as phloroglucinol derivatives, naphthodianthrones, and xanthone compounds unique to the *Hypericum* genus [2].

While the biological potential of hydrophilic molecules has been extensively studied, research on lipophilic bioactive compounds in St. John’s wort remains limited and requires further exploration [9]. The available research focuses solely on St. John’s wort leaves or the above-ground parts of the plant without distinguishing between specific sections [10]. Furthermore, studies have primarily analyzed tocopherol (T) content, with a lack of attention to tocotrienols (T3) [7]. These compounds (tocotrienols) are rare in nature, particularly in photosynthetic tissues, and their content has not been previously reported in different aerial parts of *H. perforatum* except leaves (lack of quantitative data) [11] and inflorescences [12]. Recent reports indicate a notable uniqueness of the *Hypericum* genus due to the relatively high tocotrienol content found in the leaves of seven [13] and eleven [14] species of the Hypericaceae family. This characteristic sets the *Hypericum* genus apart from other plants [13,14], including both monocots and dicots, which predominantly contain α-T in photosynthetic tissue, with tocotrienols being practically absent [15]. It is essential to explore new sources of tocotrienols since these lipophilic bioactive phytochemicals exhibit a lower abundance in nature than tocopherols. Novel discovery about the significant content of tocotrienols in St. John’s wort inflorescences [12] holds particular importance due to the extensive historical use of *H. perforatum* in traditional medicine over many centuries. For the above reasons, exploring various parts of St. John’s wort is crucial for a better understanding of the accumulation of these rare secondary metabolites. This research can also facilitate the more effective utilization of this medicinal plant. Simultaneously, it contributes to the development of sustainable use of plant materials, fundamental plant biology knowledge, and advancements in economic, pharmaceutical, and medical fields.

Tocopherols, particularly α and γ homologues, are widely distributed in nature in leaves, seeds, and plant oils. They are classified as members of the vitamin E family. However, as highlighted by Azzi [16], the term “vitamin E” should not be used interchangeably for all tocochromanol compounds. This distinction arises because only α-T fulfills the specific criteria for preventing vitamin E deficiency-related ataxia, a neurological disorder caused by insufficient vitamin E levels in humans. Tocochromanols exist in multiple forms, each varying in its antioxidant and health effectiveness [17]. α-T protects cells from oxidative stress, supports the cardiovascular system by lowering LDL levels and preventing atherosclerosis, and enhances neurological functions by slowing neurodegenerative processes. It also exhibits anti-cancer properties by stabilizing biological membranes and inhibiting the growth of certain cancers [18]. In contrast, tocotrienols and other tocochromanol-related compounds are less prevalent and have been less extensively studied and identified in natural sources [19,20,21]. As a result, there is a limited number of studies available regarding the detection, screening, taxonomic distribution, biosynthesis, metabolism, and physiological roles of these tocochromanols, despite their promising potential in promoting human health [21]. Tocotrienols lower LDL cholesterol levels, reduce inflammation, and have strong chemo—preventive effects by inhibiting the proliferation of cancer cells [18]. Tocotrienols are recognized for their safety and beneficial effects on health, as they do not pose toxicity risks to healthy cells. These compounds demonstrate anti-metastatic and anti-angiogenic properties, with a unique capacity to target and eradicate cancer stem cells selectively. Consequently, there is a growing consensus on the potential benefits of incorporating tocotrienols into mono-targeted or combination therapies in conjunction with other chemotherapeutic agents [22]. While α-T has not shown clear cancer-preventive effects, the potential impact of γ-T, δ-T, and tocotrienols on cancer risk is still unclear and requires further exploration through additional research studies [23]. Due to the positive effects of tocotrienols on human health and the occurrence of δ-T3 in nature limited to latex *Hevea brasiliensis* Muli. Arg., fruits of *Elaeis guineensis* Jacq., and seeds of *Bixa orellana* L. [24], the presence of tocotrienols in the *H. perforatum* can make this plant uniquely suitable as raw material for tocotrienol recovery in temperate climate zones.

Hence, the objective of this study is to show that various aerial parts of *H. perforatum* (stems, leaves, flower buds, flowers, dead petals, and unripe seed pods) contain relatively high concentrations of tocotrienols, especially δ-T3, tocotrienol which occur very rarely in nature, particularly in photosynthetic tissue. Knowledge demonstrated in this study can expand the use of *H. perforatum* in the food, pharmaceutical, and medical industries, as well as improve the understanding of the positive health potential of St. John’s wort.

## 2. Results and Discussion

The harvested *H. perforatum* plant and its separated parts can be seen in Appendix A. The obtained chromatograms of the profile of tocochromanols in St. John’s wort show a noticeably different composition and/or concentration in each of the analyzed parts of this plant, with some similarities (Figure 1).

The flower buds have the most diverse tocochromanol profile, while the stems have the least. Despite some differences, each part was characterized by the distinct presence of δ-T3 and α-T, constituting a substantial part of the identified tocochromanols. The exception was seed pods, where, instead of α-T, the γ-T was the main tocopherol. Figure 1 gives an illusion of domination of δ-T3 in each plant organ (the highest peak); however, due to the physicochemical properties of various tocochromanols, as well as elution order in isocratic separation by RP-HPLC-FLD, tocopherols and tocotrienols peak areas in chromatograms, especially between the homologues α and δ, are not representative of their content in the analyzed material [20]. The real content and ratio of tocopherols and tocotrienols in stems, leaves, flower buds, flowers, dead petals, and seed pods of wild *H. perforatum* can be seen in Appendix A. In four of the six investigated aerial parts of St. John’s wort—stems, leaves, flowers, and dead petals—the tocochromanol profile was similar, but their content greatly varied between the different parts. The flower buds and unripe seed pods were more diverse in terms of the tocochromanol composition than other parts of *H. perforatum*, due to the presence of considerable amounts of α-T3 (23%) and γ-T (25%), respectively. Figure 2 presents the median value, upper and lower quartile, and upper and lower extreme measurements of value distribution, while Figure 3 presents the average content proportion (%) of tocochromanols in the aerial parts of wild *H. perforatum*.

A diverse array of concentrations for both tocopherol and tocotrienol homologues, along with significant variability, were noted among the sampled aerial components of St. John’s wort (Figure 2). Both figures (Figure 2 and Figure 3) clearly demonstrate the domination of two tocochromanols, α-T and δ-T3, and concentration considerably differed between the different parts of St. John’s wort (2.1–28.2 and 2.5–37.9 mg/100 g dw, on average, respectively). Exceptions are flower buds and seed pods, which, apart from α-T and δ-T3, had significant amounts of α-T3 and γ-T (17.5–29.9 and 12.1–16.7 mg/100 g dw), respectively. In stems, leaves, flowers, and dead petals, the α-T and δ-T3 comprise 81–94% of total tocochromanols. Except for leaves dominated by tocopherols (55% of α-T), other investigated parts of St. John’s wort were predominated by tocotrienols (52–72%), where δ-T3 was a main tocochromanol (34–69%) (Figure 3). A high content of γ-T in unripe seed pods of *H. perforatum* is not surprising, since γ-T in seeds and their oil across various taxa is a predominant tocochromanol [19]. In contrast to high levels of δ-T3 and relatively notable amounts of α-T3, tocotrienol isomers (β-T3 and γ-T3) were recorded in low quantities (0.0–1.1 and 0.0–5.3 mg/100 g dw, respectively). β-T3 was present only in two parts, mainly in flower buds and sequentially in flowers. γ-T3 was detected in all *H. perforatum* parts, with the highest content in flower buds (2.7 mg/100 g dw on average). In contrast, in *Borago officinalis* L. flower buds and flowers, β-T3 and γ-T3 were the only detected tocotrienols; however, their quantities (sum of tocotrienols) were lower than recorded in the present study in *H. perforatum* flower buds and flowers [25]. From four tocopherols, three of them–β-T, γ-T, and δ-T–constituted a minor proportion of the tocochromanols, with the exception of seed pods, which contained a notable amount of γ-T (14.6 mg/100 g dw on average). The accumulation of γ-T in unripe seed pods results from seed formation. γ-T is the dominant tocochromanol in ripe *H. perforatum* seeds. Total tocochromanols content differs depending on the part of the plant. The richest source of tocochromanols were flower buds, 96.9 mg/100 g dw. On the other hand, the lowest amount was obtained for stems, 5.3 mg/100 g dw. The ratio between tocopherols and tocotrienols also differed. The biggest participation of tocopherols was for St. John wort stems, leaves, and flowers (ratio Ts/T3s–0.9) Appendix A. In most parts of a plant, α-T predominated among tocopherols, besides seed pods, where γ-T was the primary form and δ-T3 among tocotrienols (Figure 3). Tocopherols can be found in both the subterranean and aerial parts of higher plants, including roots, tubers, stems, leaves, flower buds, flowers, fruits, and seeds [24]. The predominance of α-T in St. John’s wort photosynthesis tissues (leaves and stems) is unsurprising, as α-T is well-established as the primary tocochromanol in the foliage of numerous plant species, with some exceptions where γ-T was detected as a major tocochromanol, e.g., *Kalanchoe daigremontiana* and lettuce [26]. The age of the leaf and its location play an important role in the profile of tocochromanols, as well as their concentration [26,27]. Conversely, tocotrienols are less commonly found in green plant tissues [15]. Exceptions include the species *Hypericum* and *Clusia*, where substantial quantities of tocotrienols have been identified in the leaves of five *Clusia* species and seven *Hypericum* species in one study [13], and eleven in another [14]. The composition and concentration of tocopherols and tocotrienols in the leaves of *H. perforatum* observed in this study are consistent with previous findings [13]. Among the various *Hypericum* species analyzed, *H. perforatum* uniquely demonstrates a predominance of the δ homologue among the four tocotrienols, while other *Hypericum* species primarily contain α-T3 or γ-T3 as their main tocotrienols. Interestingly, four out of the five *Clusia* species also exhibited a dominance of δ-T3 as the principal tocotrienol, similar to *H. perforatum*. It is important to note that in all *Clusia* and *Hypericum* species examined, α-T was the tocochromanol present in the highest concentrations [13,14].

The concentrations of α-T and δ-T3, two tocochromanols identified in all examined aerial parts of *H. perforatum*, exhibited significant variability in stems (coefficient of variation 0.871 and 0.471, respectively). In contrast, the lowest variability was observed in flower buds (0.122 and 0.079, respectively) Appendix A. The lowest proportion for α-T was recorded in seed pods (3%) and the highest in leaves (55%), while δ-T3 was most predominant in dead petals and seed pods (69% and 66%, respectively) and the lowest percentage was found in flower buds (34%). The highest content of tocotrienols was noted in flower buds (59.3 mg/100 g dw on average) and then subsequently in seed pods, flowers, dead petals, leaves, and stems (39.0 > 35.8 > 24.1 > 14.7 > 2.8 mg/100 g dw, respectively). Relative to other aerial parts, flower buds were characterized by a high content of α-T3 (22.7 mg/100 g dw on average), while in the other parts, α-T3 constituted much less (<5.3 mg/100 g dw). β-T3 and γ-T3 constituted the smallest portion of tocotrienols in *H. perforatum* parts and only reached notable concentration in flower buds (1.1 and 5.3 mg/100 g dw, respectively, on average). This makes δ-T3 the most dominant tocotrienol in St. John’s wort Appendix A. The prevalence of δ-T3 dominance is rare in plants, with few exceptions, such as annatto *(Bixa orellana* L.) [28] and lychee (*Litchi chinensis* Sonn.) [29] seeds and their oils, which are known for being abundant sources of δ-T3. Due to the profile and concentration of tocochromanols, St. John’s wort is more similar to lychee than to annatto seeds (tocotrienol pair domination δ-T3 and α-T3 vs. δ-T3 andγ-T3, respectively).

The composition of tocotrienols in flower buds varied the most and changed during plant development (flower buds > flowers > unripe seed pods). For three tocotrienol homologues, α-, β-, and γ-T3 content decreased in subsequent stages of development. However, δ-T3 content slightly decreased and became the lowest in flowers, and then increased to be the highest in unripe seed pods. Flowers, compared to flower buds, recorded notable losses of both α homologues (α-T and α-T3), with a particular emphasis on the α-T3. The significant decrease in α-T3 content implies the protective function of the reproductive/pollen system of the plants in St. John’s wort. This finding cannot be confirmed with literature due to a lack of studies and plant sources with high content of tocotrienols, as similarly discovered in *H. perforatum* in the present study. However, this phenomenon can be understood through several key mechanisms and findings from recent studies. One of the explanations for decreasing α-T3 content during plant development is the antioxidant/protective character of α-T3 in *H. perforatum* as a plant responds to external stress, e.g., solar radiation, no longer producing and recycling α-T3. Typically, α-T predominates as a tocochromanol in flowers, with tocotrienols present in minor quantities (Fernandes et al., 2020). α-T serves as a crucial antioxidant within chloroplasts, effectively scavenging singlet oxygen and preventing lipid peroxidation propagation, a unique function not shared by other antioxidants. This pivotal role underscores its significance in safeguarding plants against photo-oxidative damage induced by abiotic stress. Recent studies have unveiled novel roles of α-T at the systemic level, including suggesting potential non-antioxidant functions in regulating flower and fruit development as well as leaf senescence [30]. Our studies suggest that α-T3 may play a similar role in *H. perforatum*. However, this phenomenon requires future investigations for a detailed understanding. The concentration of α-T is highly variable, fluctuating in response to environmental stressors and reflecting the species’ ability to withstand such stress [31]. α-T is the main tocochromanol in photosynthetic tissues, where it is responsible for the neutralization of lipid peroxy radicals and maintaining the fluidity and integrity of membranes. The highest content of this tocochromanol was detected in flower buds, leaves, and flowers (28.2, 27.3 and 24.2 mg/100 g dw, respectively). In the other parts of a plant, the content of α-T was much lower, even 10 times lower in unripe seed pods.

Hence, the occurrence of δ-T3 in approximately 40% of the identified tocochromanols in *H. perforatum* leaves is distinctive. In the realm of scientific research, a singular instance has been documented in which photosynthetic tissues exhibited significant levels of tocotrienols, albeit at a concentration 2.5 times lower than that of tocopherols. Specifically, β and γ isomers of tocotrienol were observed in *Vellozia gigantea* leaves, which is a monocot and not related to *Hypericum* (dicot) [32]. Tocotrienols can accumulate in leaves through genetic modification, leading to a shift from a predominance of tocopherols to tocotrienols. For example, in wild-type *Nicotiana tabacum* L., α-T is the primary tocochromanol found in young leaves, but transgenic tobacco plants exhibited a higher abundance of α-T3 and γ-T3 in their leaves [33]. Nevertheless, to date, significant δ-T3 content has not been reported in the leaves of any species; however, presence of δ-T3 in *H. perforatum* leaves was already reported over decade ago (lack of quantitative data) [11]. Tocotrienols are reported mainly in latex, e.g., *H. brasiliensis*; fruits, e.g., *E. guineensis*; seeds, e.g., *B. orellana* [24]; grain bran, e.g., *Secale cereale* L. [34]; some seed oils, e.g., American cranberry (*Vaccinium macrocarpon* Aiton) and grapes (*Vitis* sp.); and cereal bran oils, e.g., wheat (*Triticum aestivum* L.) [19], but not in the leaves of wild-type plants.

Principal component analysis (PCA) was employed to analyze the data and facilitate the discovery of hidden patterns and relationships between variables. By transforming the data into new, uncorrelated variables (principal components), it becomes easier to understand the structures within the data and their interactions. The obtained results of the principal components (PC), PC1 (54.9%) and PC2 (26.3%) of the PCA explain 81.2% of the variation. PC1 was highly positively correlated (*r* ≥ 0.80) with four tocochromanols (β-T, α-T3, β-T3, and γ-T3) and moderately correlated with α-T, δ-T, and δ-T3 (*r* ≥ 0.64), whereas PC2 showed high loads only with γ-T (correlations *r* = 0.92) and was moderately correlated with δ-T and δ-T3 (*r* ≥ 0.64). According to PCA, the investigated *H. perforatum* samples were classified into six nearly completely separate groups, each group representing a different aerial part of St. John’s wort (Figure 4). The aerial parts of St. John’s wort have been separated based on the distinct profile and/or concentration of tocochromanols found in each part of the *H. perforatum* plant, as has been described above. Based on Figure 5, it can be seen that the aerial parts of *H. perforatum* differ statistically significantly (*p* < 0.05) in their concentrations of all tocochromanols.

Most noteworthy are statistical differences recorded for the content of α-T and δ-T3, the main tocochromanols in all parts of St. John’s wort. For α-T, it is possible to distinguish two groups that differ statistically significantly, while within-groups do not. The first group with low α-T content includes stems, dead petals, and seed pods. The second group with high content includes leaves, flower buds, and flowers. In the case of the δ-T3 content, most of the aerial parts of *H. perforatum* differed statistically significantly except for only two pairs, that is, leaves and dead petals, and flowers and flower buds, which were not statistically significantly different.

## 3. Materials and Methods

### 3.1. Reagents

Ethanol, methanol, ethyl acetate, *n*-hexane (HPLC grade), pyrogallol, sodium chloride, and potassium hydroxide (reagent grade) were purchased from Sigma-Aldrich (Steinheim, Germany). Ethanol (96.2%) for leaf sample saponification was received from SIA Kalsnavas Elevators (Jaunkalsnava, Latvia). Standards of tocopherol homologues (α, β, γ, and δ) (>98%, HPLC) were obtained from Extrasynthese (Genay, France), while tocotrienol homologues (α, β, γ, and δ) (>98%, HPLC) were from Cayman Chemical (Ann Arbor, MI, USA).

### 3.2. Plant Material

Nine various populations of wild *H. perforatum* were collected at the turn of June−July of 2022, during three days, from several locations in Poland (Poznań 52°25′44.5″ N 16°52′57.8″ E, Śrem-52°04′29.1″ N 17°00′09.1″ E, and Krotoszyn-51°41′52.2″ N 17°27′28.2″ E), located within a 50–100 km radius of each other. In each location, three populations of St. John’s wort located at least 0.5 km from each other were harvested. Wild *H. perforatum* was identified according to a taxonomic guide https://powo.science.kew.org (accessed on 10 May 2022). Additionally, the wild St. John’s Wort plant was visually compared with cultivated specimens grown in the Institute of Horticulture from seeds sourced from the Botanical Garden of Medicinal Plants, affiliated with the Department of Biology and Pharmaceutical Botany at Wrocław University of Medicine, located in Wrocław, Poland. The harvested wild *H. perforatum* and its aerial parts is presented on Appendix A. *H. perforatum* was collected about 2–4 weeks after full bloom when flower buds, flowers, and unripe seed pods are simultaneously present on the same plant. The intention was to enable the examination of as many different aerial parts of the plant as possible, as well as various stages of development (flower buds > flowers > seed pods). Each of the nine St. John’s wort populations was collected randomly (15–30 plants) by cutting 5–10 cm from the soil and separating into six aerial parts (stems, leaves, flower buds, flowers, dead petals, and unripe seed pods). Separate parts were pre-dried in opened small paper boxes at room temperatures +23 ± 3 °C for 10 ± 3 days, during car transport and until the final destination point (laboratory). To remove the residues of water, plant material was frozen at −80 ± 2 °C for 2 h and freeze–dried using a FreeZone freeze-dry system (Labconco, Kansas City, MO, USA) at a temperature of −51 ± 1 °C under vacuum of below 0.01 mbar for 48 h. Freeze-dried aerial parts (1–10 g each) were powdered using an MM 400 mixer mill (Retsch, Haan, Germany) and directly used for sample preparation (saponification protocol) according to the method described below. The remnants of powdered samples were transferred into polypropylene bags and stored at −18 ± 2 °C. The dry mass was measured gravimetrically.

### 3.3. Saponification and n-Hexane:ethyl Acetate Extraction Protocol

Saponification protocol was performed as described earlier [35]. The amount of 0.1 g powdered leaf sample was placed in a 15 mL glass tube with a screw cap. Then, 0.05 g of pyrogallol was added to prevent oxidation of tocopherols and tocotrienols. The mixture was sequentially supplemented with 2.5 mL of 96.2% ethanol and mixed. The process of saponification was incited by adding 0.25 mL of 60% (*w*/*v*) aqueous potassium hydroxide. The glass tube was immediately closed with a screw cap, mixed for 10–15 s using vortex REAX top (Heidolph, Schwabach Germany) with vibration frequency rates up to 2500 rpm and sequentially subjected to incubation in a water bath at 80 °C. After 10 min of incubation, the sample was mixed again for 10–15 s using vortex REAX top at 2500 rpm. After 25 min of incubation to stop/slow down the process of saponification, the sample was cooled immediately in an ice-water bath for 10 min. The process of tocopherol and tocotrienol homologues extraction started from adding 2.5 mL of 1% (*w*/*v*) sodium chloride to the glass tube with the sample to lower the surface tension between the two non-miscible solvents (hydro ethanol and *n*-hexane:ethyl acetate), and was mixed for 5 s using vortex REAX top at 2500 rpm. Then 2.5 mL of *n*-hexane:ethyl acetate (9:1; *v*/*v*) was added to extract the tocopherol and tocotrienol homologues, and mixed for 15 s using vortex REAX top at 2500 rpm. After mixing with the organic solvent mixture (*n*-hexane and ethyl acetate), the sample was centrifuged for 5 min (1000× *g*, at 4 °C). The organic layer, containing *n*-hexane and ethyl acetate, was moved to a 100 mL round bottom flask. The extraction residues were re-extracted in a fresh portion of 2.5 mL of *n*-hexane:ethyl acetate (9:1; *v*/*v*) as described above. Re-extraction was performed two times. The organic layers from initial extraction and the two re-extractions were collected and combined in the same 100 mL round-bottom flask, and evaporated in a vacuum rotary evaporator Laborota 4000 (Heidolph, Schwabach, Germany) at 40 °C until fully dry. The obtained thin film layer on the bottom of the flask was dissolved in 1 mL ethanol (HPLC grade) and transferred to 2 mL analytical glass vial.

### 3.4. Tocopherol and Tocotrienol Determination by RP-HPLC-FLD

The tocochromanol analysis were performed using reverse-phase high performance liquid chromatography with a fluorescent light detector (RP-HPLC-FLD) via HPLC Shimadzu Nexera 40 Series system (Kyoto, Japan) consisting of a pump (LC-40D pump), a degasser (DGU-405), a system controller (CBM-40), an auto injector (SIL-40C), a column oven (CTO-40C), a fluorescence detector (RF-20Axs). The chromatographic separation of tocopherol and tocotrienol homologues was carried out on the Epic PFP-LB (pentafluorophenyl phase) column (PerkinElmer, Waltham, MA, USA) with the following parameters: particle morphology, fully porous; particle size, 3 µm; column length, 150 mm; and column ID, 4.6 mm; secured with a guard column of the length 4 mm and ID 3 mm (Phenomenex, Torrance, CA, USA). The chromatography analysis was performed under the isocratic conditions as follows: mobile phase, methanol with water (91:9; *v*/*v*); flow rate, 1.0 mL/min; column oven temperature, 45 ± 1 °C; room temperature, 21 ± 1 °C. The total chromatography runtime was 13 min. The identification and quantification were performed using a fluorescence detector at an excitation wavelength of 295 nm and emission wavelength of 330 nm. The quantification was done based on the calibration curves obtained from tocopherol and tocotrienol standards.

### 3.5. Statistical Analysis

Each aerial part––stems, leaves, flower buds, flowers, dead petals, and unripe seed pods––was represented by nine biological replicates of *H. perforatum* and for all samples only one experiment was performed (6 × 9 × 1 = 54). The results for each aerial part of St. John’s wort were presented as means (*n* = 9) of the nine different population samples of *H. perforatum*. The median value, upper and lower quartile, and upper and lower extreme measurements for analysed plant material were determined using Excel (Version 2302) Microsoft 365 Apps for enterprise (Redmond, WA, USA) software. Multivariate statistical analysis and principal component analysis (PCA) of the obtained values for each sample and the eight tocochromanols (variables) (*n* = 8) was performed. To determine significant differences (α < 0.05) for identified groups by the PCA, ANOVA and post-hoc Tukey tests were applied using Statistica 13.0 (TIBCO Software Inc., Palo Alto, CA, USA).

## 4. Conclusions

This study provides a comprehensive analysis of the tocochromanol profile in various aerial parts of *H. perforatum* (St. John’s wort), revealing significant diversity in composition and concentration across plant tissues. Key findings highlight δ-T3 as the dominant tocotrienol in all aerial components, with α-T prevailing among tocopherols, particularly in photosynthetic tissues like leaves and stems. Except the leaves, dominant in α-T, δ-T3 was the main tocochromanol in the aerial parts of St. John’s wort. Flower buds emerged as the richest source of tocochromanols, exhibiting a diverse composition of tocotrienols, while the unripe seed pods were characterized by the highest content of δ-T3. *H. perforatum* may be recognized as a valuable source of δ-T3 in temperate climate zones. Knowledge demonstrated in this study can expand the use of *H. perforatum* in the food, pharmaceutical, and medical industries as well as improve the understanding of the positive health potential of St. John’s wort.

The precise selection of plant parts from St. John’s wort can optimize the production of biologically active compounds while minimizing waste and environmental impact due to the multifaceted use of this medicinal plant. Future studies should focus on developing sustainable extraction techniques and optimizing purification processes to obtain high-quality bioactive products. Ultimately, conducting clinical trials may be essential to maximize the therapeutic potential of this medicinal plant, particularly concerning the interactions among its various phytochemicals, including δ-T3.

## Figures and Tables

**Figure 1 molecules-30-01137-f001:**
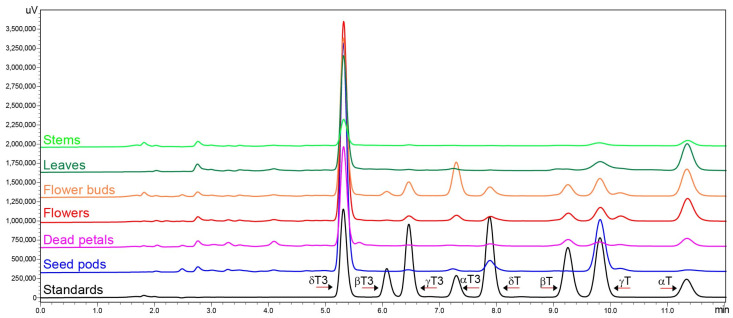
Chromatograms of the tocotrienol (T3) and tocopherol (T) homologues (α, β, γ, and δ) separation by RP-HPLC/FLD in wild *H. perforatum* stems, leaves, flower buds, flowers, dead petals, seed pods, and standards.

**Figure 2 molecules-30-01137-f002:**
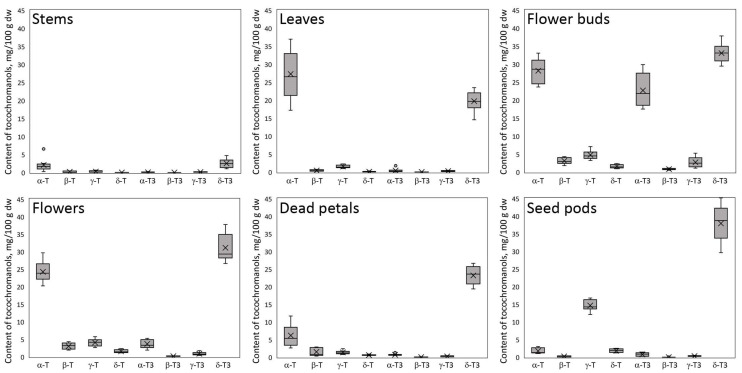
The box-plot of the content of tocotrienol (T3) and tocopherol (T) homologues (α, β, γ, and δ) in stems, leaves, flower buds, flowers, dead petals, and seed pods of wild *H. perforatum*.

**Figure 3 molecules-30-01137-f003:**
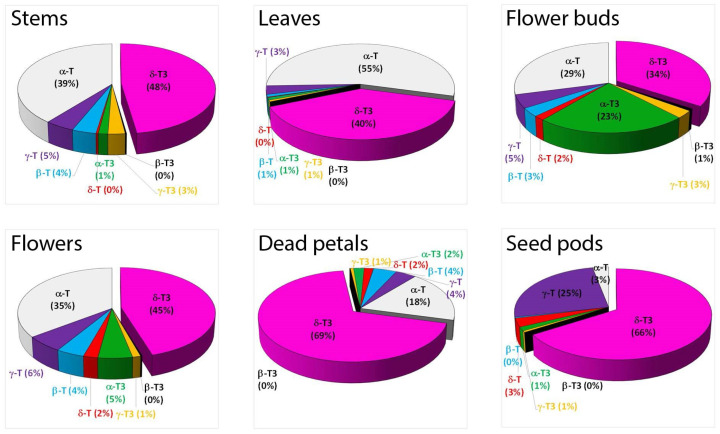
The average content proportion (%) of individual tocotrienol (T3) and tocopherol (T) homologues (α, β, γ, and δ) in stems, leaves, flower buds, flowers, dead petals, and seed pods of wild *H. perforatum*.

**Figure 4 molecules-30-01137-f004:**
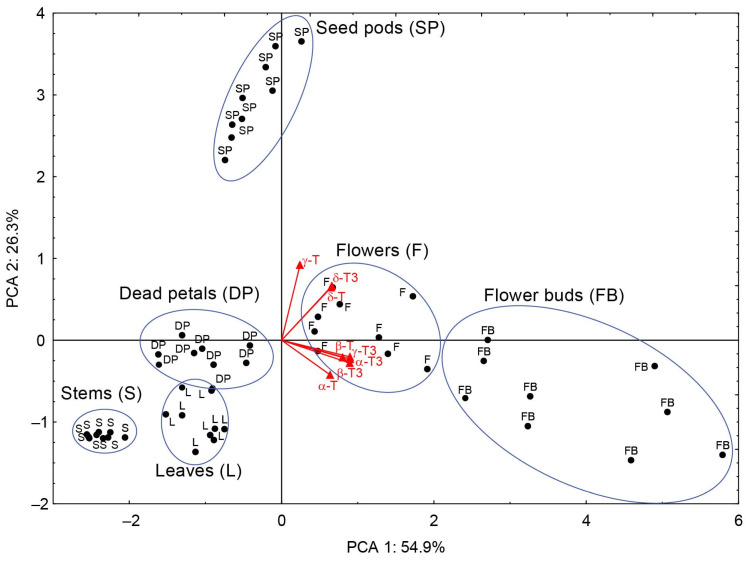
Principle component analysis (PCA), with variability-factor coordinates plot and case-factor coordinates plot, according to the content of tocotrienol (T3) and tocopherol (T) homologues (α, β, γ, and δ) in six aerial parts of *H. perforatum*.

**Figure 5 molecules-30-01137-f005:**
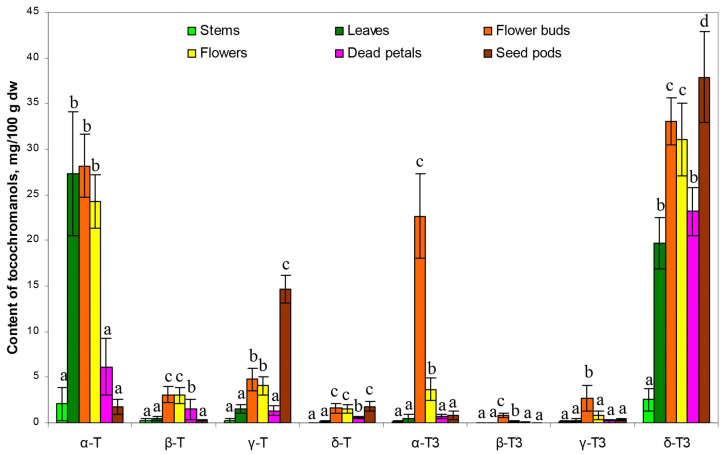
The content of tocotrienol (T3) and tocopherol (T) homologues (α, β, γ, and δ) in groups (stems, leaves, flower buds, flowers, dead petals, and seed pods) assigned by the PCA. Statistically significant differences among groups were assigned by ANOVA and Tukey test (*p* < 0.05), where different letters denote significant differences between groups.

## Data Availability

The data used to support the findings of this study are available in Appendix A and from the corresponding author upon request.

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
