# Peer review of "Tocotrienols in Different Parts of Wild Hypericum perforatum L. Populations in Poland"

_molecules, 2025, doi:10.3390/molecules30051137_

Round 1

Reviewer 1 Report

Comments and Suggestions for Authors

The manuscript has analyzed the tocotrienols in different parts of wild Hypericum perforatum populations in Poland. The contents were clearly stated, afford to the relevant professional field. The manuscript was scientifically sound, and its experimental design was appropriated for detecting questions raised by the author. The data of the manuscript was sound, and the results were reproducible; the conclusions were consistent with the evidence and arguments presented. Therefore, this manuscript is worth publishing.

Author Response

We sincerely thank you for all the comments, remarks, and suggestions that have contributed to enhancing the manuscript and its scientific quality. The manuscript have been improved accordingly. Provided changes are marked in red font. For literature we used references manager software therefore changes are not highlighted.

Reviewer 1

Comments 1: The manuscript has analyzed the tocotrienols in different parts of wild Hypericum perforatum populations in Poland. The contents were clearly stated, afford to the relevant professional field. The manuscript was scientifically sound, and its experimental design was appropriated for detecting questions raised by the author. The data of the manuscript was sound, and the results were reproducible; the conclusions were consistent with the evidence and arguments presented. Therefore, this manuscript is worth publishing.

Response 1: Thank you for your positive feedback and for taking the time to review our manuscript. We truly appreciate your insightful comments and are pleased to hear that you find our study scientifically sound and suitable for publication.

Reviewer 2 Report

Comments and Suggestions for Authors

This manuscript reports the quantitative analysis of tocotrienol and tocopherol contents in the different parts of St. John's wort using HPLC. In this manuscript, the analysis results are only reported descriptively, and I think that it lacks significant scientific importance. It is natural that the content of any substance differs depending on the part of the plant, and I believe that simply reporting these differences is insufficient as a scientific research.

Author Response

We sincerely thank you for all the comments, remarks, and suggestions that have contributed to enhancing the manuscript and its scientific quality. The manuscript have been improved accordingly. Provided changes are marked in red font. For literature we used references manager software therefore changes are not highlighted.

Reviewer 2

Comments 1: This manuscript reports the quantitative analysis of tocotrienol and tocopherol contents in the different parts of St. John's wort using HPLC. In this manuscript, the analysis results are only reported descriptively, and I think that it lacks significant scientific importance. It is natural that the content of any substance differs depending on the part of the plant, and I believe that simply reporting these differences is insufficient as a scientific research.

Response 1: Thank you for the comment. Our study provides unique insights into the tocochromanol profile in different parts of H. perforatum. Yes, it is obvious that the phytochemical composition and concentration will differ in different parts, but the question is not if it will differ but how. Why it is important? H. perforatum is studied for several decades due to its unique metabolites, but the information about the presence of a relatively high concentration of tocotrienols is reported only presently. Green parts of different plant species contain mainly α-T as the main tocochromanol, while tocotrienols are found very rarely or mostly in trace amounts. Additionally, studies show that H. perforatum is dominated by δ-T3, which is one of the rarest tocotrienols after β-T3. This makes it worth taking a closer look at this plant in terms of the content of these rare bioactive compounds in its different parts. To our knowledge, the distribution of these compounds across various parts of the plant has not been previously studied, and our findings provide novel data that can be useful for future research and applications of this medical plant. We made several improvements and explanations to highlight the uniqueness and novelty of the present manuscript. (Page 2, Page 5, bottom part, Page 9, top part, Page 10, bottom part)

Reviewer 3 Report

Comments and Suggestions for Authors

The manuscript "Tocotrienols in different parts of wild Hypericum perforatum L. populations in Poland" presents data that are undoubtedly of interest to the scientific community. However, despite the presence of well-prepared graphic material and a broad descriptive section, there are some comments on the manuscript:

- L 79: Why was this designation introduced - (vitamin E = α-T)? In the introduction, the abbreviations for tocopherols and tocotrienols should be introduced at the first mention, as readers without the relevant knowledge may misunderstand the information presented.

- L 405-406: the reference is incorrect.

- Overall the manuscript is well written, but most of it is taken up by the description of the results, presented in several graphs.

- The authors conclude that the use of H. perforatum is expanding in the food, pharmaceutical and medical industries. It is unclear whether the authors are talking about the potential use of H. perforatum itself or that it can be used as a source for the extraction of biologically active substances, including tocotrienols. If used as a source for extraction of biologically active substances, have studies been conducted to select methods for extraction and purification of BAS?

Author Response

We sincerely thank you for all the comments, remarks, and suggestions that have contributed to enhancing the manuscript and its scientific quality. The manuscript have been improved accordingly. Provided changes are marked in red font. For literature we used references manager software therefore changes are not highlighted.

Reviewer 3

Comments 1: The manuscript “Tocotrienols in different parts of wild Hypericum perforatum L. populations in Poland” presents data that are undoubtedly of interest to the scientific community. However, despite the presence of well-prepared graphic material and a broad descriptive section, there are some comments on the manuscript:

Response 1: Thank you for the positive overview.

Comments 2: - L 79: Why was this designation introduced - (vitamin E = α-T)? In the introduction, the abbreviations for tocopherols and tocotrienols should be introduced at the first mention, as readers without the relevant knowledge may misunderstand the information presented.

Response 2: Thank you for the comment. The text has been changed accordingly:

Original text: They are a class of compounds within the vitamin E family (vitamin E = α-T), and are recognized for their significant health benefits due to their antioxidant and an-ti-inflammatory properties.

Changed text: “They are classified as members of the vitamin E family. However, as highlighted by Azzi [13], the term "vitamin E" should not be used interchangeably for all tocochromanol compounds. This distinction arises because only α-T fulfills the specific criteria for preventing vitamin E deficiency-related ataxia, a neurological disorder caused by insufficient vitamin E levels in humans.” (Page 2, bottom part)

To first mention about tocopherols and tocotrienols in line 68 the abbreviations were added accordingly:

Original text: Furthermore, the studies primarily analyzed tocopherol content, with a lack of attention to tocotrienols [7]. These compounds (tocotrienols) are rare in nature…

Changed text: Furthermore, the studies primarily analyzed tocopherol (T) content, with a lack of attention to tocotrienols (T3) [7]. These compounds (tocotrienols) are rare in nature… (Page 2, bottom part)

Comments 3: - L 405-406: the reference is incorrect.

Response 3: Thank you for noticing. The reference has been changed accordingly:

“Linde, Klaus, St. John's wort ‒ an overview. Forschende Komplementärmedizin/Research in Complementary Medicine 2009, 16, (3), 146-155.” (Page 11, top part)

Comments 4: - Overall the manuscript is well written, but most of it is taken up by the description of the results, presented in several graphs.

Response 4: Thank you for the comment. Regarding the emphasis on result descriptions and graphical presentations, our goal was to provide a clear and comprehensive visualization of the data to enhance the reader's understanding. Several topics were discussed in more detail in the present version of the manuscript. (Page 5, bottom part)

Comments 5: - The authors conclude that the use of H. perforatum is expanding in the food, pharmaceutical and medical industries. It is unclear whether the authors are talking about the potential use of H. perforatum itself or that it can be used as a source for the extraction of biologically active substances, including tocotrienols. If used as a source for extraction of biologically active substances, have studies been conducted to select methods for extraction and purification of BAS?

Response 5: Thank you for the comment. The conclusion part has been improved accordingly: “The precise selection of plant parts from St. John's wort can optimize the production of biologically active compounds while minimizing waste and environmental impact due to the multifaceted use of this medicinal plant. Future studies should focus on developing sustainable extraction techniques and optimizing purification processes to obtain high-quality bioactive products. Ultimately, conducting clinical trials may be essential to maximize the therapeutic potential of this medicinal plant, particularly concerning the interactions among its various phytochemicals, including δ-T3.” (Page 10, bottom part)

Round 2

Reviewer 2 Report

Comments and Suggestions for Authors

The author adds some explanation of the importance of this research, but my opinion remains unchanged. If you are going to discuss the differences in the production of metabolic products in different parts of a plant, it is not enough to simply report the differences; you should also consider the physiological or ecological significance of the differences and the mechanisms that cause them (you will need to do experiments to provide evidence).